# Psychophysiological Effects of *Lactobacillus plantarum* PS128 in Patients with Major Depressive Disorder: A Preliminary 8-Week Open Trial

**DOI:** 10.3390/nu13113731

**Published:** 2021-10-22

**Authors:** Hui-Mei Chen, Po-Hsiu Kuo, Chia-Yueh Hsu, Yi-Hung Chiu, Yen-Wenn Liu, Mong-Liang Lu, Chun-Hsin Chen

**Affiliations:** 1Institute of Epidemiology and Preventive Medicine, National Taiwan University, Taipei 100, Taiwan; hmchen2019@gmail.com; 2Department of Psychiatry, National Taiwan University Hospital, Taipei 100, Taiwan; 3Department of Psychiatry, Taipei Municipal Wan-Fang Hospital, Taipei Medical University, Taipei 116, Taiwan; joanne.cy.hsu@gmail.com (C.-Y.H.); chiuyihang@gmail.com (Y.-H.C.); mongliang@hotmail.com (M.-L.L.); 4Department of Psychiatry, School of Medicine, College of Medicine, Taipei Medical University, Taipei 110, Taiwan; 5Institute of Biochemistry of Molecular Biology, National Yang-Ming University, Taipei 112, Taiwan; skywenn@gmail.com

**Keywords:** probiotics, major depressive disorder, inflammation, gut permeability, microbiota

## Abstract

Recent studies have suggested that gut–brain axis may be one of the mechanisms of major depression disorder (MDD). The current study aimed to investigate the effects of *Lactobacillus plantarum PS128* (PS128) on psychophysiology in patients with MDD. We recruited 11 patients with MDD and gave them PS128 for 8 weeks. We compared depression symptoms, serum markers of inflammation and gut permeability, and gut microbiota before and after 8-week intervention and also explored the correlations among symptoms, biomarkers, and gut microbiota. After 8-week PS128 intervention, scores of Hamilton Depression Rating Scale-17 and Depression and Somatic symptoms Scale significantly decreased. Serum levels of high sensitivity c-reactive protein, interluekin-6, and tumor necrosis factor-α, zonulin and intestinal fatty acid binding protein, and the composition of gut microbiota did not significantly change after 8-week PS128 intervention. However, we found changes of some genera were correlated with changes of symptoms and biomarkers. In conclusion, this is an open trial with small sample size and has several limitations. The results need to be verified by randomized, double-blind, placebo-controlled trial with larger sample size.

## 1. Introduction

Major depressive disorder (MDD) is a complex, long-term illness and exhibits marked disabilities in affected patients [1]. Treatment generally needs to be taken for at least three months before the symptoms of MDD might be alleviated. Thus, patients with depressive disorders can endure more gastrointestinal problems than healthy controls [2]. In addition, each patient responds differently to the medication [3]. Recently, probiotics have been prescribed to alleviate the symptoms of gastrointestinal problems; however, few relative studies focus on patients with MDD. As the intestinal epithelia are partly regulated by the gut microbiota, which is tightly linking the immune systems of the host, it is essential to understand what degree of impact consuming probiotic has on patients with MDD.

Inflammation and depression are intertwined. According to the results of meta-analyses, the levels of inflammation indications, including interleukin-6 (IL-6), tumor necrosis factor-α (TNF-α), and C-reactive protein (CRP), were significantly higher in MDD patients [4,5]. A systemic inflammation within patients with MDD could be contributed by bacterial translocation due to leaky gut [6]. In addition, the findings of higher levels of IgM and IgA against gram-negative microbiota in patients with MDD than those in normal volunteers suggest that increased gut permeability could increase translocation of gram-negative microbiota in patients with MDD [7]. Zonulin [8] and intestinal fatty acid binding protein (I-FABP) [9] have been used as indicators of gut permeability. Their variances remarkably differed between patients with depression or anxiety disorders and controls [10]. It is also noted that patients who had recently attempted suicide had a significantly high level of serum I-FABP but a low level of zonulin compared with patients who had not attempted suicide and healthy controls [11]. Nevertheless, there are still no clear insights to explain the potential involvement of gut bacteria linking between inflammation and depression.

The recognition of the value for probiotics for human health can be traced back in the early 1900s [12]. Nowadays, the effects of probiotic interventions have been shown in different fields of medicine, such as protection against diarrheal diseases, lowering of cholesterol, and stimulation of the immune system [13,14,15,16]. Additionally, early studies have been conducted to evaluate the applicability in dentistry, where they have shown a significant improvement of periodontal diseases [17]. Nonetheless, the mechanisms of probiotics for health effects are still unclear. Some studies have suggested that the mechanisms of the probiotics activity are possibly via direct or indirect actions between the gut microbiome and the intestinal immune system.

The hypothesis that probiotics could be adjuvant treatment in MDD patients was proposed first in 2005 due to its potential to lower systemic inflammatory cytokine, decrease oxidative stress, and improve nutritional status [18]. Accordingly, psychobiotics were defined in 2013 based on the results of sufficient preclinical studies of probiotics such as *Bifidobacterium* and *Lactobacillus* [19,20]. Increased gut permeability could increase translocation of gram-negative microbiota in patients with MDD [7]. In contrary, *Bifidobaterium infants* are able to enhance epithelial cell barrier function in numerous, diverse ways, aiding in developing and maintaining homeostasis [21]. There have been several clinical trials to use probiotics to alleviate depressive symptoms, and meta-analysis results show that probiotics are effective only in patients with MDD but not in non-MDD population [22].

*Lactobacillus plantarum* PS128 (PS128) is one of bacteria extracted from traditional fermented food, Fu-Tsai [23]. It can improve anxiety-like behavior and increase monoamine neurotransmitters in the striatum of germ-free mice [24]. It also alleviates depressive-like behavior and reduces inflammation level in mice with early-life stress [25]. Inspired by the early preclinical studies of PS128, we considered that there might be changes of gut microbiota, markers of inflammation, and gut permeability observed following a probiotic PS128 intervention for eight weeks. Thus, the null hypotheses examined in the present study for PS128 intervention were (1) there is no change of depressive symptoms, (2) the markers of inflammation and gut permeability do not change, and (3) there is no difference in terms of microbial diversity.

## 2. Materials and Methods

### 2.1. Study Participants

The inclusion criteria required participants who were adults, whose ages ranged from 20 to 65. All participants fulfilled the Diagnostic and Statistical Manual of Mental Disorders fifth version (DSM-5) criteria of MDD in the past two years. Additionally, any psychotropics prescribed to the participants, including antidepressants, antipsychotics, and hypnotics, had remained unchanged for at least one month prior to the start of the study. All participants’ HAMD score, Hamilton Depression Rating Scale-17 items (HAMD-17) [26], were greater or equal to 14.

The exclusion criteria were as follows: (1) comorbid with DSM-5 diagnoses of schizophrenia, bipolar disorder, or combined with substance use (except tobacco) disorder; (2) having active suicidal or homicidal ideation; (3) known allergy to probiotics; (4) comorbid with diabetes mellitus, irritable bowel syndrome, inflammatory bowel disease, liver cirrhosis, or autoimmune diseases; (5) known active bacterial, fungal, or viral infections within the past month; (6) use of antibiotics, steroid, immunosuppressants, or probiotics in the month before collecting blood and fecal samples; (7) women who have ever been pregnant or lactating women; (8) recent, strong changes to dietary pattern or diet within the month prior to the commencement of the study.

### 2.2. Study Design

We gave recruited patients one PS128 capsule twice a day: one in the morning and one in the afternoon. Each PS128 capsule contains 300 mg of probiotics, equivalent to 3 × 10^10^ CFU of *Lactobacillus plantarum* PS128. PS128 was manufactured and packed by BENED Biomedical Co. Ltd. (Taipei, Taiwan). As activity and diet may alter composition of gut microbiota, we asked eligible patients not to change their lifestyle or dietary pattern. In addition, though effects of psychotropics on inflammation are still controversial [27], medications were kept unchanged during the intervention period. If psychotropic were changed because of worsening clinical condition, the patient was withdrawn from the study. Adherence was monitored by pill counts at each visit.

Recruited patients were visited at baseline and weeks 2, 4, and 8. Depression assessment was evaluated at every visit. We drew blood samples to test markers of inflammation and gut permeability at baseline and week 8. Fecal samples for analyzing the composition of gut microbiota were collected at baseline and week 8. Approximately 2 g of fecal samples were collected at home each time and delivered at 4 °C, transferred to 2 c.c eppendorfs, and immediately stored at −80 °C for further analysis.

### 2.3. Ethical Consideration

This 8-week, open-label trial was approved by the Institutional Review Board of Taipei Medical University before recruiting patients, and the study was conducted in accordance with the approval guidelines (N201804031). Written informed consent was obtained from every participant and the implications of the trial had.

### 2.4. Depression Assessment

HAMD-17 [26] and Depression and Somatic symptoms Scale (DSSS) [28] were used to assess patients’ depressive severity. HAMD-17 contains 17 items to rate the severity of mood, feeling of guilt, suicide ideation, insomnia, agitation, anxiety, weight loss, and somatic changes. The DSSS is a self-administered scale and composed of 22 items with two major subscales, the depression subscale (DS) and the somatic subscale (SS). The DS had 12 items, including three vegetative symptoms and fatigue, and the SS had 10 items, including five pain items, which comprised the pain subscale (PS) [28].

### 2.5. Markers of Inflammation

Serum levels of high sensitivity CRP (hs-CRP), TNF-α receptor, and IL-6 receptor were analyzed using commercial enzyme-linked immunosorbent assay (ELISA) kits according to the assay protocol provided by the supplier (R&D systems, Minneapolis, MN, USA). The final absorbance of each sample for the mixture was measured and analyzed at 450 nm using an ELISA plate reader with Bio-Tek Power Wave Xs and BioTek Gen 5 (version 3.11, Biotek, Winooski, VT, USA). ELISA has been considered the gold standard of immunoassays [29].

### 2.6. Markers of Gut Permeability

Zonulin is a protein identified as prehaptoglobulin-2 (the precursor of haptoglobin-2) and a regulator of intestinal permeability [30]. Serum zonulin concentration was analyzed by using commercial kits (R&D systems, Minneapolis, MN, USA) according to the manufacturer’s instructions. I-FABP is found in the enterocytes of the small intestine, and elevated levels indicate enterocyte damage [31]. Serum I-FABP was measured using a sandwich enzyme immunoassay (R&D systems, Minneapolis, MN, USA) according to the manufacturer’s instructions.

### 2.7. Fecal Sample Collection

Fecal samples were self-collected by participants at home and were put in refrigerator before delivering to laboratory. Approximately 2 g of the fecal samples were collected within 2 weeks of the interview. All fecal samples were delivered in 4 °C and stored in −80 °C refrigerator immediately after transferring to 2 c.c eppendorf. The protocol of fecal sample collection was as previously described [32].

### 2.8. 16S rRNA Gene Amplification Sequences

Bacterial DNA was isolated from fecal samples using the bead-beating method as previously described [32]. The V3-V4 regions of the 16S rRNA gene were amplified in PCR using the following forward primer (5-CCTACGGGNGGCWGCAG-3) and reverse primer (5-GACTACHVGGGTATCTAATCC-3). The 16S rRNA gene samples were sequenced on the Illumina MiSeq platform with 300 bp paired-end reads. Library construction and sequencing were performed at the First Core Laboratory, National Taiwan University. The 16S rRNA gene sequences from the paired Fastq files were processed using the QIIME2 pipeline in combination with Deblur denoising and closed reference clustering [33,34]. Taxonomy was assigned using the Greengenes database [35].

### 2.9. Genera Selection Process for Correlation Heatmaps

Twenty-one genera were selected for further analysis according to the following filter and analysis processes. First, the genera that were detected in at least 50% of all fecal samples were selected. Second, the differences between baseline and week 8 were calculated in three separate datasets: the relative abundance of genera, the biomarker levels, and the questionnaire scores. The differences in datasets were denoted by ΔG for genera, ΔB for biomarkers, and ΔQ for questionnaires. Third, correlations of difference (Δ) between the three separate datasets were evaluated using the Spearman correlation coefficient with the Wilcoxon rank-sum test. Fourth, correlated pairs with an absolute value of correlation coefficient higher than 0.5 and *p*-values less than 0.05 were selected. Finally, genera that appeared in more than one significant pair were only included once within the final Venn diagram. The analytical process is shown in Appendix A.

### 2.10. Statistical Analysis

We used descriptive statistics to display demographic and clinical data in the recruited patients. Repeated measure of analysis of variance was used to test the changes of depressive symptoms among baseline, week 2, week 4, and week 8. Paired *t*-test was used to test the changes of markers of inflammation and gut permeability between baseline and week 8. Differences in the relative abundance of phyla and genera, alpha diversity, and beta diversity were evaluated by the Wilcoxon rank-sum test. The principal coordinate analysis method was used for visualization of beta diversity. The correlation matrix was generated by the rcorr() function from Hmisc package in R. The heatmaps of the correlations were generated in R. R software was used for statistical analysis (The R Project for Statistical Computing, Vienna, Austria).

## 3. Results

### 3.1. Changes of Clinical Measurements and Serum Biomarkers

A total of 11 patients with MDD, eight women and three men, were recruited and completed the 8-week trial. Among them, five patients used escitalopram, one sertraline, two vortioxetine, one duloxetine, one agomelatine, and one without any medication. The mean age (standard deviation, SD) was 39.4 (12.0) years. Table 1 shows the changes of depressive severity, and the markers of inflammation and gut permeability, during 8-week PS128 intervention. After 8-week PS128 intervention, HAMD-17, DSSS, and subscales of DSSS were significantly improved, whereas the markers of inflammation and gut permeability did not change remarkably. In addition, the change in biomarkers did not significantly correlate with the change in severity of symptoms measured by HAMD-17 and DSSS. However, the change in pain subscale (DSSS-PS) was positively correlated with all five biomarkers (Appendix A).

### 3.2. Differences in the Microbiota Composition of the Study Population

The composition of gut microbiota was analyzed in 18 fecal samples from 9 patients at two time points. At the phylum level, Firmicutes dominated within almost all fecal samples, followed by Bacteroidetes and Actinobacteria (Figure 1A, Appendix A). No phyla differed significantly between the baseline and after eight weeks of intervention (Appendix A).

At the genus level, a total of 113 genera were detected within the whole fecal sample, and the dominant genera differed between fecal samples. Figure 1B shows the microbiota composition of the average relative abundance of the 14 top genera within the whole fecal sample. The alpha diversity, measured by richness and the Shannon diversity index, and the beta diversity, measured by the Bray–Curtis distance, did not change significantly after the consumption of probiotics (Figure 2). Nevertheless, six genera—*Akermensia*, *Bifidobacterium*, *Enterococcus*, *Eggerthella*, *Megasphaera* and (*Ruminococcus*)—changed significantly in terms of the rank of the difference in relative abundance between the two time points or in their *p*-values (Appendix A).

### 3.3. The Degree of Microbial Co-Occurrence within Specific Genera Changed after Consumption of Probiotics

To explore connections in microbial communities, two microbial co-occurrence networks of 21 specific genera were constructed using the Spearman’s correlation (Figure 3). Within the 210 genera pairs, 12 genera pairs maintained high levels of correlation. Two genera, *Ruminococcus* and *Odoribacter,* strongly correlated with eight other genera at the baseline and after eight weeks, respectively, which was the highest number of genera according to the results of correlation coefficients of 210 genera pairs. In addition, *Ruminococcus* were all positively correlated with the eight genera *Anaerostipes*, *Coprococcus*, *Gemmiger*, *Oscillospira*, *Paraprevotella*, *Roseburia*, *Slackia*, and *Veillonella*. In contrast, 76 genera pairs changed their relationship from positive correlation to negative correlation or vice versa (Appendix A). Nevertheless, within the 76 genera pairs, the correlation coefficient between *Akkermansia* and *Prevotella* changed with moderate significance, and only three genera pairs seemed to change remarkably, which were *Roseburia* and *Anaerostipes*, *Veillonella* and *Gemmiger*, and *Veillonella* and *Ruminococcus*.

### 3.4. The Association among Changes of Symptoms, Serum Biomarkers, and Gut Microbiota

Fifty genera that were detected in at least 50% of all fecal samples were selected for further analysis of their associations with changes of biomarkers and symptoms severity, respectively. The differences between baseline and week 8 were calculated in three separate datasets: the relative abundance of genera, the biomarker levels, and the questionnaire scores. Figure 4 depicts their relationships using heatmaps. Of these, seven genera significantly correlated with IL-6 (*Akkermansia*, *Coprococcus*, *Paraprevotella*, *Phascolarctobacterium*, *Prevotella*, *Oscillospira,* and *Turicibacter)*; five genera with TNF-alpha (*Anaerostipes*, *Enterococcus*, *Lactobacillus*, *Gemmiger*, and *Oscillospira*); three genera with I-FABP (*Phascolarctobacterium*, *Pseudomonas*, and *Turicibacter*); and one genus (*cc_115,* of family Erysipelotrichaceae) with zonulin (Appendix A).

It is noted that the values of the subclass DSSS-PS scores, measuring the degree of pain, were significantly correlated with *Slackia*, with a correlation of approximately 0.93 and *p*-value less than 0.001. In addition, DSSS positively correlated with two genera, *Bifidobacterium* and *Lactobacillus,* and negatively correlated with *Sutterella* (Appendix A). Only two genera, *Coprococcus* and *Lactobacillus,* significantly correlated with both biomarkers and depressive symptoms. Figure 5 depicts their relationships using heatmaps.

## 4. Discussion

According to our results, the null hypothesis about changes of depressive symptoms was rejected, while the other two were not rejected, i.e., the serum levels of inflammation markers and the most dominant bacteria were barely influenced after eight weeks of consuming PS128 in the within individual comparisons. However, two microbial correlation networks within specific genera illustrated some differences over the intervention. To the best of our knowledge, it is the first study to investigate the changes of biomarkers of gut permeability and gut microbiota after probiotics intervention in patients with MDD.

The effects of probiotics on markers of inflammation have been reported in preclinical studies, yet the results are inconsistent in human, especially in limited clinical studies. An early study showed that serum level of hs-CRP significantly decreased after 8 weeks of *Lactobacillus acidophilus*, *Lactobacillus casei,* and *Bifidobacterium bifidum* in patients with MDD, and the depressive symptoms of the patients significantly decreased [36]. In contrast, a recent study found no significant changes in depression severity and inflammatory markers after 8-week intervention of *Lactobacillus helveticus* and *Bifidobacterium longum* in participants with depressive symptoms [37]. Until now, it is still challenging to compare across previous study findings due to the differences in methodology, such as the characteristics of participants and the strains of probiotics used. Nevertheless, according to the results of meta-analysis of the effects of probiotics on depressive symptoms, probiotics are effective in ameliorating depressive symptoms in patients with MDD [22]. The effects of probiotics on inflammation markers in patients with MDD still needs to be elucidated.

Gut permeability also plays a vital role in the communication between gut and brain [38]. In our study, we did not find significant changes of the mean levels of I-FABP and zonulin between baseline and 8-week PS128 intervention. Furthermore, the microbial diversity did not change significantly. Because microbial communities in fecal samples for adults were generally stable over a period of months except for rare events, such as enteric infection or consuming antibiotics [39], the phenomenon of minor changes of the microbial diversity could be attributable to the fact that participants in our study maintained their lifestyle and diet habit and that the probiotic consumption was only eight weeks.

The biomarker levels and microbial abundance only revealed a small degree of variation in our patients’ fecal samples. Some genera showed potential correlations with biomarkers. In our study, according to the results of the associations between the differences of biomarkers and the differences of genera abundances, genus *Akkermansia* and the level of IL-6 is strongly correlated. *Akkermansia* is in the phylum Verrucomicrobia and has also been reported to be associated with the proinflammatory pathways [40]. A recent study proved that the relative abundance of *Akkermansia spp*. was significantly declined in socially defeated animals, which displayed an increase in depressive-like behavior [41]. We also found that the mean relative abundance of *Akkermansia* was significantly higher in MDD patients after consuming probiotics than at baseline in the study.

There are some limitations in the current pilot study. First, the diet information of participants was lacking. Nonetheless, the participants were asked to maintain their dietary habit during intervention. Second, the small sample size, 11 patients with MDD, was limited the reliability of the results and could amplify minor effects of probiotics on markers of inflammation and gut permeability. Third, the period of observation was only 8 weeks. The duration may not have been long enough to detect changes in markers of gut permeability and the composition of gut microbiota. According to our previous meta-analysis of the effects of probiotics on depression [22], the duration of the intervention of probiotics in patients with MDD ranges from 6 to 8 weeks. These studies did not analyze markers of gut permeability and gut microbiota. Therefore, we do not know whether the intervention of 8 weeks is long enough to change markers of gut permeability and the composition of gut microbiota. It could be better to design multiple collection time points and to extend the study for more than three months. Finally, the study was an open trial and lacks a control group.

In conclusion, we found that depressive severity in patients with MDD significantly ameliorated, but markers of inflammation, gut permeability, and the composition of gut microbiota did not significantly change after 8 weeks of PS128 intervention. Whether PS128 is effective in patients with MDD needs to be verified using a randomized, double-blinded, placebo-controlled study with larger sample size in the future.

## Figures and Tables

**Figure 1 nutrients-13-03731-f001:**
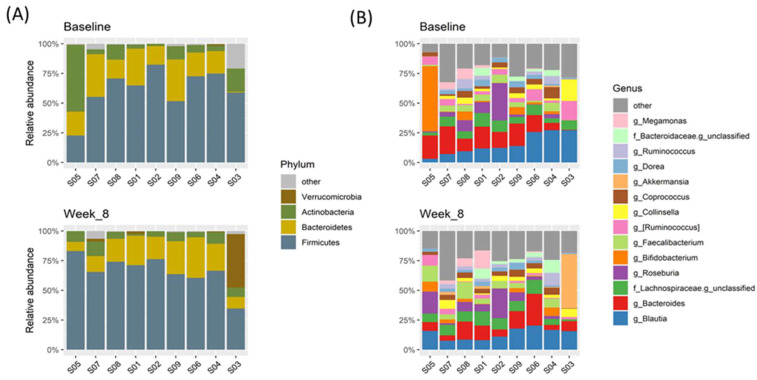
Microbiota composition at phylum and genus levels of each fecal sample at the two collection times. (**A**) Firmicutes was the most dominant phylum within almost all fecal samples. The differences of microbial abundance did not change significantly between the two sample collection times. (**B**) Almost all fecal samples were dominated by *Blautia* and *Bacteroides*, except one sample. Sample ID S05 was dominated by *Bifidobacterium*, of which relative abundance exceeded 50% at baseline but reduced to less than 15% at eight weeks.

**Figure 2 nutrients-13-03731-f002:**
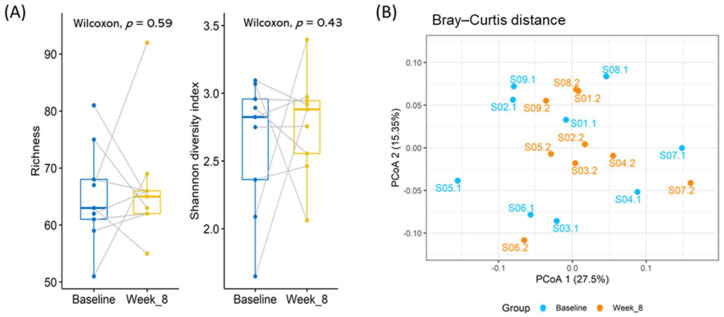
Microbial diversity at genus level. (**A**) Alpha diversity was assessed using richness and the Shannon diversity index. The richness of microbial diversity means the number of genera detected within each fecal sample. Both richness and the Shannon diversity index did not show significant differences between the two sample collection times. (**B**) Principal coordinate analysis (PCoA) was based on the Bray–Curtis distance. Each point represents one fecal sample. The two colors represent the two collection times, with blue denoting baseline and orange denoting eight weeks after consumption of PS128.

**Figure 3 nutrients-13-03731-f003:**
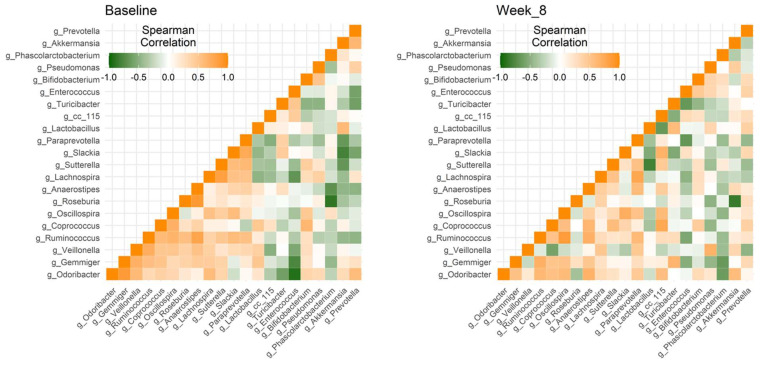
Microbial co-occurrence networks at genus level. The two heat maps represent microbial co-occurrence at the two sample collection times within 21 genera that were detected in at least 50% of all 18 fecal samples. The strength was measured using Spearman’s correlation. In general, the patterns of those microbial correlations are somewhat different between the baseline and at the end of trial.

**Figure 4 nutrients-13-03731-f004:**
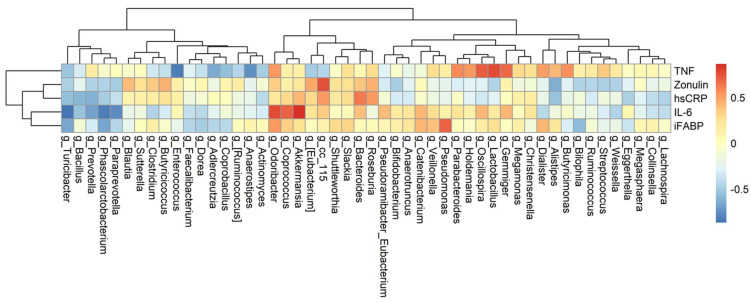
Correlation heatmap between genera and biomarkers. Fifty genera were detected in at least 50% of all 18 fecal samples. The differences in value of 50 genera and that of five biomarkers between two collection times were calculated first and then evaluated for their strength of correlation using Spearman’s correlation. Red denotes positive correlation, and blue denotes negative correlation. Column names of the heatmaps are the genera. The values of five biomarkers (TNF, zonulin, hs-CRP, IL-6, and I-FABP) and the relative abundance of genera were transformed using the logarithm function. Abbreviations: hsCRP, high sensitivity C-reactive protein; iFABP, intestinal fatty acid binding protein; IL-6, interleukin-6; TNF, tumor necrosis factor-α.

**Figure 5 nutrients-13-03731-f005:**
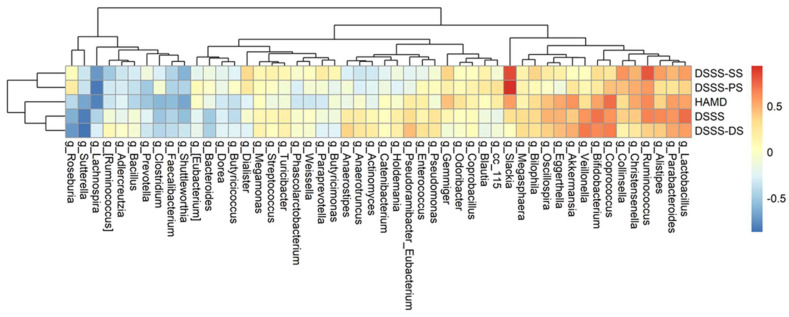
Correlation heatmaps between genera and the questionnaire scores. Fifty genera were detected in at least 50% of all 18 fecal samples. The differences in value of 50 genera and that of five biomarkers between two collection times were calculated first and then evaluated for their strength of correlation using Spearman’s correlation. Red denotes positive correlation, and blue denotes negative correlation. Column names of the heatmaps are the genera. The row names are the severity symptom measurements HAMD-17 and DSSS including the subscales of DSSS. Abbreviations: DSSS, Depression and Somatic symptoms Scale; HAMD-17, Hamilton Depression Rating Scale-17 items.

**Table 1 nutrients-13-03731-t001:** Changes of depression and markers of inflammation and gut permeability during 8-week PS-128 intervention (*n* = 11).

	Baseline	Week 2	Week 4	Week 8	*P* ^a^
	Mean (SD)	Mean (SD)	Mean (SD)	Mean (SD)	
HAMD-17	20.1 (5.6)	12 (4.2)	16.2 (5.8)	12.0 (6.1)	0.01 ^a^
DSSS, total score	31.0 (4.5)	20.2 (3.8)	22.2 (3.6)	18.4 (3.5)	<0.001 ^a^
DSSS, depressive	19.2 (9.0)	12.6 (8.3)	13.8 (7.6)	11.1 (6.9)	<0.001 ^a^
DSSS, somatic	11.8 (6.9)	7.6 (5.6)	8.4 (5.6)	7.3 (6.6)	0.001 ^a^
DSSS, pain	5.6 (3.5)	3.7 (2.9)	4.0 (2.7)	3.5 (3.4)	0.006 ^a^
hs-CRP, *n* = 10	3.1 (3.7)	-	-	4.1 (6.5)	0.71
IL-6, ng/mL	35.7 (10.1)	-	-	35.0 (11.9)	0.61
TNF-α, pg/mL	886.9 (248.8)	-	-	961.5 (304.1)	0.13
Zonulin, mg/mL	0.94 (0.48)	-	-	0.92 (0.54)	0.92
I-FABP, pg/mL	1.67 (11.2)	-	-	1.46 (1.02)	0.71

^a^ repeated measures analysis of variance. Abbreviations: DSSS, Depression and Somatic symptoms Scale; HAMD-17, Hamilton Depression Rating Scale-17 items; hs-CRP, high sensitivity C-reactive protein; I-FABP, intestinal fatty acid binding protein; IL-6, interleukin-6; TNF-α, tumor necrosis factor-α. “-“ means that the data are not available.

## Data Availability

The data that support the findings of this study are available from the corresponding author upon reasonable request.

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
