# Peer review of "Psychophysiological Effects of Lactobacillus plantarum PS128 in Patients with Major Depressive Disorder: A Preliminary 8-Week Open Trial"

_nutrients, 2021, doi:10.3390/nu13113731_

Round 1

Reviewer 1 Report

Dear Authors,

Thank you for your work dealing with the psychophysiological effects of a probiotic in patients with major depressive disorder. I think it is very interesting from a clinical point of view.

I just have some few minor suggestion to improve the quality of your work:

  • I suggest to add the statistical null hypothesis at the end of the introduction
  • I suggest to add a reference for section 2.5, 2.7 and 2.8
  • I suggest to add in the discussion the acceptance or not of the statistical null hypothesis.
  • You should add in the introduction or discussion that probiotics are now studied in different fields of medicine and, in the last years, more and more studies have been conducted to evaluate the applicability even in dentistry where they show a significant improvement of periodontal disease. 

Congratulations for your work.

Yours faithfully

Author Response

Introductory comment:

Thank you for your work dealing with the psychophysiological effects of a probiotic in patients with major depressive disorder. I think it is very interesting from a clinical point of view. I just have some few minor suggestion to improve the quality of your work:

Response to comment: We thank the reviewer for the positive feedback and appreciate the reviewer’s careful and thorough reading this manuscript. The thoughtful comments and suggestions have helped us to improve the quality of this manuscript. We have carefully considered the comments and tried our best to address every one of them.

Comment #1:

I suggest to add the statistical null hypothesis at the end of the introduction.

Response to comment #1: We thank the reviewer for this suggestion. We have changed the last few sentences (with underline) of the Introduction section, which includes the statistical null hypotheses.

Inspired by the early preclinical studies of PS128, we considered that there might be changes of gut microbiota, markers of inflammation and gut permeability observed following a probiotic PS128 intervention for eight weeks. Thus, the null hypotheses examined in the present study for PS128 intervention were (1) there is no change of depressive symptoms, (2) the markers of inflammation and gut permeability do not change, and (3) there is no difference in terms of microbial diversity.

Comment #2:

I suggest to add a reference for section 2.5, 2.7 and 2.8

Response to comment #2: We thank the reviewer for pointing out the necessity to include the references in the three sections. We have added the references and related texts (with underline) for each of the three sections.

In section 2.5: ELISA has been considered the gold standard of immunoassays [29].

In section 2.7: The protocol of fecal sample collection was as previously described [32].

In section 2.8: The 16S rRNA gene sequences from the paired Fastq files were processed using the QIIME2 pipeline in combination with Deblur denoising and closed reference clustering [33, 34]. Taxonomy was assigned using the Greengenes database [35].

References:

29. Alhajj M, Farhana A: Enzyme Linked Immunosorbent Assay. In: StatPearls. Treasure Island (FL); 2021.

32. Chung YE, Chen HC, Chou HL, Chen IM, Lee MS, Chuang LC, Liu YW, Lu ML, Chen CH, Wu CS et al: Exploration of microbiota targets for major depressive disorder and mood related traits. J Psychiatr Res 2019, 111:74-82.

33. Bolyen E, Rideout JR, Dillon MR, Bokulich NA, Abnet CC, Al-Ghalith GA, Alexander H, Alm EJ, Arumugam M, Asnicar F et al: Reproducible, interactive, scalable and extensible microbiome data science using QIIME 2. Nat Biotechnol 2019, 37(8):852-857.

34. Amir A, McDonald D, Navas-Molina JA, Kopylova E, Morton JT, Zech Xu Z, Kightley EP, Thompson LR, Hyde ER, Gonzalez A et al: Deblur Rapidly Resolves Single-Nucleotide Community Sequence Patterns. mSystems 2017, 2(2).

35. DeSantis TZ, Hugenholtz P, Larsen N, Rojas M, Brodie EL, Keller K, Huber T, Dalevi D, Hu P, Andersen GL: Greengenes, a chimera-checked 16S rRNA gene database and workbench compatible with ARB. Appl Environ Microbiol 2006, 72(7):5069-5072.

Comment #3:

I suggest to add in the discussion the acceptance or not of the statistical null hypothesis.

Response to comment #3: We thank the reviewer for this suggestion. We have added the sentence (with underline) in the first paragraph of the discussion section.

“According to our results, the null hypothesis about changes of depressive symptoms was rejected, while the other two were not rejected, i.e. the serum levels of inflammation markers and the most dominant bacteria were barely influenced after eight weeks of consuming PS128 in the within individual comparisons.”

Comment #4:

You should add in the introduction or discussion that probiotics are now studied in different fields of medicine and, in the last years, more and more studies have been conducted to evaluate the applicability even in dentistry where they show a significant improvement of periodontal disease

Response to comment #4: We are grateful for this comment as it points to the importance of probiotics in improving different treatments for human diseases not only focusing on the mental disorders in this study. We have added the description (with underline) in the third paragraph of the introduction section.

The recognition of the value for probiotics for human health can be traced back in the early 1900s [12]. Nowadays, the effects of probiotic interventions have been shown in different fields of medicine, such as protection against diarrheal diseases, lowering of cholesterol and the stimulation of the immune system [13-16]. Additionally, early studies have been conducted to evaluate the applicability in dentistry where they have shown a significant improvement of periodontal diseases [17]. Nonetheless, the mechanisms of probiotics for health effects are still unclear. Some studies have suggested that the mechanisms of the activity are possibly via direct or indirect actions between the gut microbiome and the intestinal immune system.

Reviewer 2 Report

To investigate the effects of Lactobacillus planta- 15 rum PS128 (PS128) on psychophysiology in patients with MDD, the author examined depression symptoms, serum markers of inflammation and gut permeability, and gut microbiota before and after 8-week intervention in 11 patients with MDD. The results showed that after 8-week PS128 intervention, scores of Hamilton Depression Rating Scale-17 significantly decreased, but that serum levels of high sensitivity c-reactive protein, interluekin-6, and tumor necrosis factor-alpha, the composition of gut microbiota did not significantly change in 10 patients with MDD.

Although the study has been performed systematically, there are some concerns regarding the method, the results, and the discussion.

  • The sample size with 11 cases are too small to analyze the change in some biomarkers including the composition of gut microbiota after 8-week PS128 intervention. The lack of change in the markers of inflammation and gut permeability seems to be due to a small sample size.
  • The duration of 8 weeks may be not enough long to affect the gut permeability and gut microbiota. In fact, no phyla differed significantly between the baseline and after eight weeks intervention.

  • In the discussion (line 307) the author described that the phenomenon of minor changes of the microbial diversity could be attributable to the participants in our study maintained their lifestyle and diet habit, and that the consuming probiotics was only eight weeks.
    Indeed no change in life style and diet habits might account for no difference in the microbial diversity after intervention. However, the inter-individuals variation of the microbial diversity might be due to the variation of drug therapy among individuals.  The author should discuss the effects of the variation of drug use among participants.

Author Response

Comment #1:

The sample size with 11 cases are too small to analyze the change in some biomarkers including the composition of gut microbiota after 8-week PS128 intervention. The lack of change in the markers of inflammation and gut permeability seems to be due to a small sample size.

 Response to comment #1: Small sample size indeed is our main limitation for detecting the changes of markers of inflammation and gut permeability. This is a pilot study and we provided the first line evidence for the preliminary results in this field. We have written in our limitation. Moreover, we intend to conduct a future study using a randomized, double-blinded, placebo-controlled study with a larger sample size to validate the current findings.

Comment #2:

The duration of 8 weeks may be not enough long to affect the gut permeability and gut microbiota. In fact, no phyla differed significantly between the baseline and after eight weeks intervention.

 Response to comment #2: Thanks for your comment. The duration of 8 weeks may be not long enough to affect gut permeability and the composition of gut microbiota. According to our previous meta-analysis of the effects of probiotics on depression, the duration of the intervention of probiotics in patients with MDD ranges from 6 to 8 weeks. These studies did not analyze markers of gut permeability and gut microbiota. We have added it into our limitation and discussed more.

“…the period of observation was only 8 weeks. The duration may not have been long enough to detect changes in markers of gut permeability and the composition of gut microbiota. According to our previous meta-analysis of the effects of probiotics on depression [22], the duration of the intervention of probiotics in patients with MDD ranges from 6 to 8 weeks. These studies did not analyze markers of gut permeability and gut microbiota. Therefore, we do not know whether the intervention of 8 weeks is long enough to change markers of gut permeability and the composition of gut microbiota.”

Comment #3:

In the discussion (line 307) the author described that the phenomenon of minor changes of the microbial diversity could be attributable to the participants in our study maintained their lifestyle and diet habit, and that the consuming probiotics was only eight weeks.

 Indeed no change in life style and diet habits might account for no difference in the microbial diversity after intervention. However, the inter-individual variation of the microbial diversity might be due to the variation of drug therapy among individuals. The author should discuss the effects of the variation of drug use among participants.

 Response to comment #3: We added medications which participants used in the results on page 9. Five patients used escitalopram, 1 sertraline, 2 vortioxetine, 1 duloxetine, 1 agomelatine and 1 without any medication. Because the sample size is limited, we did not stratify subjects based on their medication types to compare their microbial diversity.

…Among them, five patients used escitalopram, one sertraline, two vortioxetine, one duloxetine, one agomelatine, and one without any medication…

In addition, we tested the differences in markers of inflammation and gut permeability, and gut microbiome within individuals before and after 8-week intervention, and the medications during the intervention were not changed. Therefore, the changes of microbial diversity are unlikely affected by the variation of drug therapy among individuals.

Although there were no remarkable changes in the overall statistics for the biomarkers in the pilot study, we observed a significant depressive change following the consumption of probiotics on an individual level across all patients. As far as we know, currently, there has been no comparative study to investigate the changes of biomarkers of gut permeability and gut microbiota following probiotics intervention in patients with MDD. Based on the findings of this pilot study and our learnings from the way that the study was conducted, we intend to conduct a future study using a randomized, double-blinded, placebo-controlled study with a larger sample size to validate the current findings.

Round 2

Reviewer 2 Report

The manuscript has been improved well.